# INTELLIGENCE AT THE EDGE OF CHAOS

Shiyang Zhang[*,1,2]    Aakash Patel[*,1]    Syed Rizvi[1]    Nianchen Liu [3]    Sizhuang He[1]
Amin Karbasi[1]    Emanuele Zappala[4]    David van Dijk[†,1]

[1]Yale University, [2]Columbia University, [3]Northwestern University, [4]Idaho State University

## ABSTRACT

We explore the emergence of intelligent behavior in artificial systems by investigating how the complexity of rule-based systems influences the capabilities of models trained to predict these rules. Our study focuses on elementary cellular automata (ECA), simple yet powerful one-dimensional systems that generate behaviors ranging from trivial to highly complex. By training distinct Large Language Models (LLMs) on different ECAs, we evaluated the relationship between the complexity of the data generated by the rules and the models' ability to learn effective general representations, as reflected in their performance on downstream tasks. Our findings reveal that models trained on more complex data exhibit greater predictive ability, as demonstrated by their performance on reasoning and chess move prediction tasks. Both uniform and periodic systems, and often also highly chaotic systems, resulted in poorer downstream performance, highlighting a sweet spot of complexity conducive to intelligence. We conjecture that intelligence arises from the ability to predict complexity and that creating intelligence may require only exposure to complexity.

## 1 INTRODUCTION

The emergence and nature of intelligence within computational systems have long been subjects of fascination and rigorous study in the fields of artificial intelligence (AI) and theoretical computation. Traditional AI methodologies predominantly involve training models on high-quality datasets inherently imbued with human intelligence—such as natural language corpora, expert-annotated datasets, or data reflecting human cognitive processes (Coleman et al., 2019). This approach operates under the assumption that creating intelligent behavior necessitates exposure to intelligent data sources. In contrast, this paper explores an alternative hypothesis: that intelligence can emerge from modeling simple systems as long as they exhibit complex behaviors, even when the process that generates the data lacks inherent intelligence.

To investigate this hypothesis, we utilize Stephen Wolfram's elementary cellular automata (ECA) as our experimental framework. ECAs are one-dimensional, binary-state, discrete computational systems defined by 256 possible 8-bit rules. They generate a diverse spectrum of behaviors ranging from simple, repetitive patterns to highly complex and chaotic structures (Wolfram, 1983). Despite their simple rule-based definitions, certain ECAs produce significantly complex patterns, making them ideal for examining the relationship between complexity and intelligence.

Our methodology involves training separate instances of the GPT-2 language model (Radford et al., 2019) on data generated by individual ECAs. The models are tasked with predicting future states of the automata. Following this pretraining phase, we evaluate the transformer models' ability to learn useful representations by quantifying their performance on downstream logical reasoning and chess move prediction tasks.

This paper presents an extensive study exploring the relationship between system complexity and the emergence of intelligence in large language models (LLMs), quantified by the effectiveness of

---

[*]Equal contribution
[†]Correspondence to `david.vandijk@yale.edu`

learned representations for downstream tasks. We discover a positive correlation between the complexity of the ECA data and the downstream performance of models trained on that data, highlighting the role of complexity in learning effective representations. Surprisingly, we find that models can learn complex solutions even when trained on rules that generate simple data. Our results suggest an optimal complexity level, or "edge of chaos", conducive to learning, where the system is structured yet challenging to predict. These findings enhance our understanding of intelligence in artificial systems and provide a framework for future research focused on the importance of complexity in developing these systems.

## 2 BACKGROUND

### 2.1 ELEMENTARY CELLULAR AUTOMATA

Cellular automata (CAs) are computational models of complex systems, consisting of a grid of cells that evolve over time based on simple rules. First introduced by John von Neumann (von Neumann, 1966), CAs have since been widely used to simulate various physical, biological, and computational systems due to their simplicity and ability to produce complex behavior.

Elementary Cellular Automata (ECAs) (Wolfram & Mallinckrodt, 1994) are a type of one-dimensional cellular automaton where each cell has a binary state, and its next state is determined by a simple rule that depends only on the current state of the cell and its two immediate neighbors. There are 256 possible ECA rules, 88 of which are unique after accounting for symmetries (Castillo-Ramirez & Magaña-Chavez, 2023). Notable examples include Rule 110, which has been proven to be Turing complete (Cook, 2009), and Rule 90, which generates the fractal-like Sierpinski triangle. These rules are categorized into four classes based on their behavior when initialized with random conditions: Class I, which evolves to a homogeneous state; Class II, which forms simple periodic structures; Class III, which produces chaotic and aperiodic patterns; and Class IV, which exhibits complex structures (Castillo-Ramirez & Magaña-Chavez, 2023).

ECAs are valuable computational models used to explore complex systems and emergent behaviors arising from simple rules. Their usefulness lies in their simplicity and the variety of patterns they can produce, making them ideal for studying pattern formation in computation (Meunier, 2016), physics (Banerjee & Dalui, 2024), and mathematical biology (Rasolonjanahary & Vasiev, 2020). Additionally, ECAs have been utilized in cryptography as a basis of certain security frameworks (Corona-Bermúdez et al., 2022) and in computer science education (Staubitz et al., 2016) to illustrate concepts in algorithms and computational theory. Their ability to model intricate systems with minimal computational resources has made ECAs a popular tool across scientific disciplines.

### 2.2 LARGE LANGUAGE MODELS

Large language models (LLMs) are advanced artificial intelligence systems designed to understand and generate human-like text based on vast datasets (Brown et al., 2020). By leveraging deep learning techniques, these models analyze patterns in language to perform tasks such as translation, summarization, and conversational dialogue (Devlin et al., 2018). Notable examples like OpenAI's GPT-4 have demonstrated remarkable capabilities in producing coherent and contextually relevant responses across a wide range of topics (Achiam et al., 2023). The development of LLMs represents a significant advancement in natural language processing, opening up new possibilities for applications in education, research, and industry (Katz et al., 2023).

### 2.3 COMPLEXITY MEASURES

Various complexity measures have been proposed to assess the behavior of dynamical systems. In this work, we employ the following measures:

1. **Lempel-Ziv Complexity** assesses the compressibility of a sequence by counting the number of unique substrings in the sequence (Lempel & Ziv, 1976).

2. **Compression Complexity** quantifies how effectively a sequence can be compressed using a data compression algorithm such as Zlib (Zli).

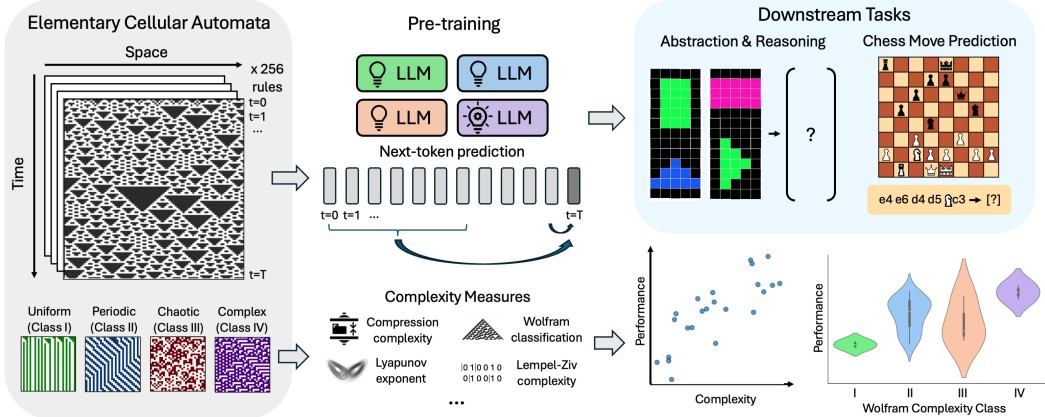

Figure 1: Our framework for investigating the link between complexity and intelligence. We pretrain Large Language Models (LLMs) on Elementary Cellular Automata (ECAs) from different complexity classes using next-token prediction, then evaluate them on downstream reasoning and chess move prediction tasks. We use various measures to analyze the complexity of ECA-generated data, and quantify the relationship between complexity and downstream performance.

3. **Lyapunov Exponent** gauges a system's sensitivity to initial conditions. Higher Lyapunov exponents indicate that small variations in initial states result in rapidly diverging outcomes. We adopt the method proposed by Wolf (1986) for computing this metric.

4. **Krylov Complexity** evaluates how information propagates in a system's Hilbert space, measuring how quickly an operator spans larger regions of the state space over time (Parker et al., 2019).

5. **Wolfram Classification** categorizes ECA rules into four categories based on behavior and complexity (see Section 2.1).

While these measures are correlated with one another, each measures different aspects of complexity. For most analyses, we focus on Lempel-Ziv Complexity and the Wolfram Classification. Performance on downstream tasks as a function of other complexity measures is shown in Appendix E.

## 3 METHODOLOGY

In this study, we systematically investigate the relationship between system complexity and the generalization of learned representations of models trained on these systems. This section outlines our methodology, including the steps for data generation and model pretraining. An overview of the training process and task evaluations is provided in Figure 1.

### 3.1 DATA GENERATION

To train our models, we simulate a selection of ECA rules. Each simulation generates a sequence of binary vectors, where each vector represents the system's spatial state at a specific time step. For each sample, we begin with a randomly initialized vector as the automaton's initial state. The system is then evolved over 1000 time steps by repeatedly applying the chosen ECA rule. This process produces a sequence of binary vectors that capture the evolving dynamics of the ECA over time.

To increase the diversity of the training data, we extract random spatiotemporal windows from the full sequences. Specifically, we sample subsequences by selecting random windows of 60 time steps and 100 spatial dimensions from the binary vectors. This method exposes the model to a variety of contexts and state configurations, enhancing its ability to learn the dynamics of the ECA rules and generalize to new sequences. Each training sequence represents a randomly selected segment in

space and time of the automaton's evolution. To vary the difficulty of the task, we train models to predict either 1 or 5 steps in the future.

## 3.2 Training Procedure for GPT-2 Models

We utilized a modified GPT-2 architecture (Radford et al., 2019) adapted for binary input and output data, enabling it to perform next-token prediction on sequences of binary vectors. Instead of using a traditional token embedding layer followed by a softmax over a vocabulary, we replaced the token embeddings with a linear projection layer that directly maps binary vectors into the model's embedding space. The GPT-2 model processes these embeddings to capture temporal dependencies and patterns within the sequences. At the output, we apply a linear projection layer to map the model's hidden states back to the data dimensionality, generating the prediction for the next state of binary variables at each time step. This adaptation allows the GPT-2 model to handle binary data directly and perform next-token prediction without relying on a predefined vocabulary. Additionally, this also makes the model deterministic, in line with the deterministic nature of ECAs.

## 3.3 Pretraining Setup

Each model was pretrained on next-token prediction tasks using data generated from a single ECA rule for up to 10,000 epochs. In each epoch, a new dataset was generated from a random initial state. Not only does this setup effectively provide infinite data, but it also makes the volume of data seen by the model directly proportional to the number of epochs the model has been trained for. To prevent overfitting, early stopping based on validation loss was employed. The training data were organized into batches of 64 sequences, each comprising 60 time steps and 100 spatial dimensions.

We employed the Adam optimizer with an initial learning rate $\eta = 2 \times 10^{-6}$ and a weight decay of 0.01. A learning rate scheduler with a linear warm-up over the first 10% of the total steps was implemented to stabilize the initial stages of training and improve convergence rates. After the warm-up phase, we applied cosine annealing to gradually decay the learning rate over the remaining training steps. Gradient accumulation was used to handle larger effective batch sizes within the constraints of GPU memory, allowing us to simulate larger batch sizes by accumulating gradients over multiple mini-batches. To prevent exploding gradients, we applied gradient clipping with a maximum norm of 1.0.

## 4 Experiments

To evaluate the emergent intelligence of models trained on cellular automata, we conducted experiments on three downstream tasks: one easy and one hard reasoning task inspired by the Abstraction and Reasoning Corpus (ARC) (Chollet, 2019), and a challenging chess move prediction task (Ruoss et al., 2024). These tasks were designed to quantify the transformer models' abilities in reasoning, abstraction, and long-term prediction, thereby assessing the level of intelligence encoded during pretraining on ECA-generated data of varying complexities. We freeze the layers of the pretrained GPT-2 models and train only the input and output projection layers for downstream tasks to ensure that performance differences reflect the inherent capabilities of the models.

Our central hypothesis is that models pretrained on rules that generate complex data will exhibit superior performance on downstream tasks compared to those pretrained on simple rules. The inclusion of both easy and hard tasks allows us to observe different performance trends and better understand the relationship between pretraining complexity, task difficulty, and emergent intelligence. The chess move prediction task, in particular, serves as an excellent system to test reasoning due to its inherent complexity and requirement for strategic thinking.

## 4.1 Downstream Task: Reasoning

We developed a downstream task inspired by the ARC (Chollet, 2019) to evaluate models' problem-solving and reasoning abilities. Our approach utilizes sequence completion problems that require the model to infer transformation rules from provided examples and apply them to novel scenarios.

The data consists of a fixed number of shapes on a grid. At each time step, any of the following transformations can be applied to each shape: changing the **color**, **rotating** the shape by $90°$, or **shifting** the shapes by one position. We designed two versions of the reasoning task, differing in complexity based on the number and type of transformations applied.

**Easy:** In the easy task, we only use $3 \times 3$ squares that are fixed in position and orientation. The only possible transformation is a color change, which we perform in a predetermined order.

**Hard:** The hard task involves more complex patterns where all transformations are applied simultaneously. We used four distinct base shapes, each represented by a $5 \times 5$ matrix. This results in complex sequences where shapes change color, orientation, and position over time. The combination of these transformations requires the model to reason over multiple simultaneous changes to accurately predict the next pattern in the sequence.

For the easy reasoning task, models were post-trained for 1,000 epochs with a learning rate of $1 \times 10^{-4}$. Due to the increased complexity and difficulty of the hard reasoning task, we post-train for 10,000 epochs with a learning rate of $1 \times 10^{-5}$. We use early stopping based on validation loss to prevent overfitting.

## 4.2 Downstream Task: Chess Move Prediction

For the chess experiment, we evaluated the capability of the different ECA-pretrained models to predict next moves in chess games represented using Standard Algebraic Notation (SAN) (Alg). We use chess games from the Lichess Elite database (Lic), focusing on games played between January and April 2016 by Grandmasters with ratings of 2200 and above. Each game was represented as a sequence of SAN moves. We split this collection into training, validation, and test sets using an 80-10-10 split to facilitate model training and evaluation. Each game sequence was segmented into subsequences of 60 moves each, and any subsequence shorter than this length was padded sequences to length 60.

We added an embedding layer to convert the SAN tokens into vector representations, which were then processed using the frozen ECA-pretrained model. A linear output layer was used to transform the outputs to the vocabulary size corresponding to the SAN tokens. These input and output layers were trained while the rest of the model was frozen. The model was trained using cross-entropy loss, the Adam optimizer (Kingma, 2014), and a learning rate scheduler with warm-up. Early stopping was again used.

## 4.3 Hardware and Software

The experiments were conducted using PyTorch version 2.1.2 and the Transformers library (version 4.41.0), with CUDA version 12.4 for GPU acceleration. The models were trained on 12 NVIDIA H100 GPUs, each with 80 GB of memory, running on Red Hat Enterprise Linux 8.8.

## 5 Results

In this section, we present our results exploring the relationship between system complexity and emergent intelligence in LLMs. The following sections detail our analyses of task performance and attention patterns across models trained on data with varying complexities.

## 5.1 Relationship between Intelligence and Complexity

Figure 2 presents the model performance across three downstream tasks (easy reasoning, hard reasoning, and chess move prediction) as a function of the complexity of the ECA rules the models were pretrained on. The top row highlights the relationship between performance and the Lempel-Ziv complexity data, while the bottom row categorizes the performance by Wolfram's complexity classes. For clarity, two representative rules from each complexity class are displayed on the left, with their corresponding performance annotated in the top plots.

For the reasoning tasks, models generally achieve near-perfect accuracy when trained for a sufficient number of epochs. Therefore, instead of reporting absolute accuracy, we focus on model efficiency,

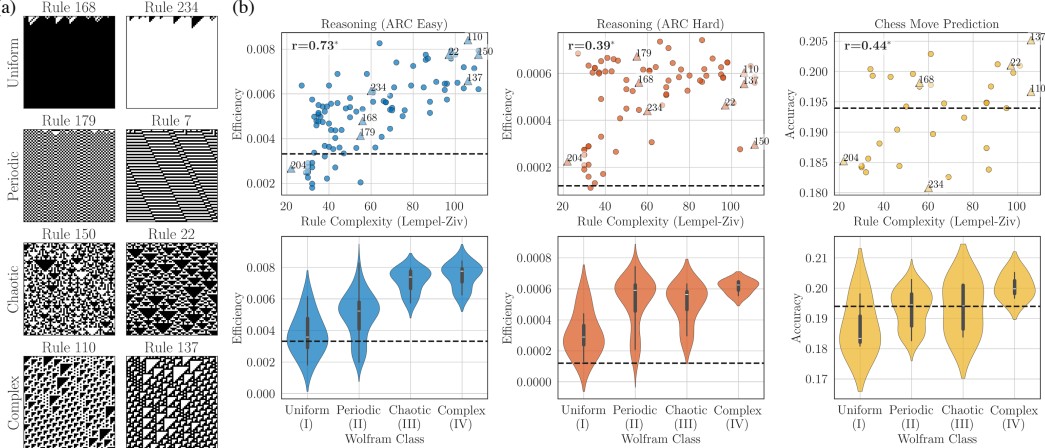

Figure 2: Relationship between downstream task performance and data complexity. **(a)** Eight representative ECA rules, two from each of Wolfram's four complexity classes. Performance of models trained on these rules is highlighted in the top row of (b). **(b)** Top row: Model performance in relation to the Lempel-Ziv complexity of data generated by each rule. The left and center panels show efficiency (1 divided by number of epochs to reach 80% validation accuracy) for the easy and hard reasoning tasks, respectively. The right panel shows move prediction accuracy for the chess task. The rules depicted on the left are highlighted in the plot with triangles and annotated with the rule number. The correlation coefficient is shown in the top-left corner of each plot. An asterisk next to the value indicates a significant relationship ($p < 0.05$). Bottom row: Downstream task performance based on Wolfram classification of each rule. Models trained on Class III and Class IV (chaotic and complex) rules perform better than models trained on uniform and simple rules. Baseline results for a randomly initialized transformer model are shown with a dashed black line on all plots.

defined as the inverse of the number of epochs required to reach 80% accuracy. The chess task is sufficiently difficult that models do not achieve perfect performance, and so we report the final accuracy. As data complexity increases, we observe a clear positive correlation in all tasks, with more complex data leading to greater efficiency. This correlation is significant for each of the tasks. The p-values, along with the distance correlation coefficient, are given in Appendix D.

In terms of Wolfram's classification, rules from Classes I and II (uniform and periodic) show lower average efficiency in the reasoning tasks compared to those from Classes III and IV (chaotic and complex). Class IV rules especially outperform the other classes on the chess move prediction task. This pattern suggests that models trained on more complex data tend to perform better on harder downstream tasks. Results with respect to other complexity measures are shown in Appendix E.

We observe that models trained on certain Class III (Chaotic) rules, such as Rules 105, 146, and 150, have poorer performance on the hard reasoning and chess move prediction tasks. This behavior is expected due to chaotic systems lacking the structured patterns necessary for effective learning. In other words, they may be too random to predict, leading to weaker downstream performance. Though we provide the correlation coefficient to quantify the relationship, the key takeaway is the qualitative insight: performance peaks at intermediate complexity and deteriorates with excessive complexity. These results highlight the existence of a "sweet spot" of complexity conducive to intelligence, where the system is still predictable yet hard to predict.

## 5.2 MODELS LEARN COMPLEX SOLUTIONS FOR SIMPLE RULES

The elementary cellular automata (ECA) are inherently *memoryless*, meaning the state at the next time point is determined only by the current time point, without any consideration of past states. For each model, a straightforward solution exists: simply learning the 8-bit ECA rule and applying it to the current state to predict the next state. However, alternative solutions may also be possible, where the model leverages historical states for its predictions. The key question is whether the model is merely learning the trivial solution or if it is integrating information from the state history.

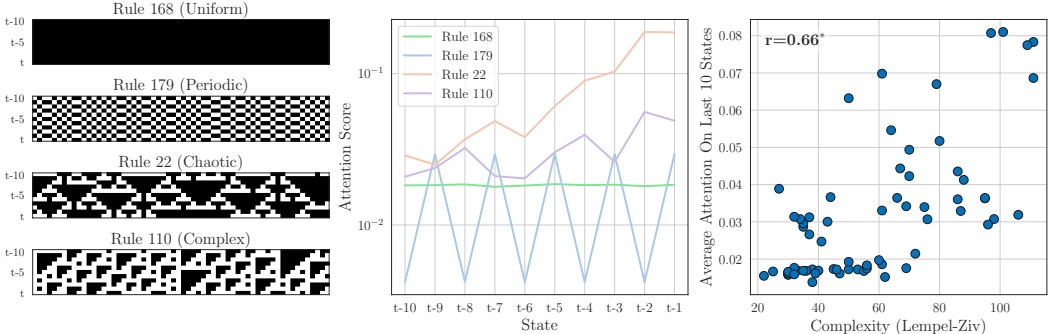

Figure 3: Attention scores for the final 10 states prior to the target state, showing that models trained on more complex data rely more heavily on past states for prediction. **Left:** Visualization of the last 10 states and the target state for representative rules from each of Wolfram's complexity classes. **Center:** Attention scores for each of the last 10 states, highlighting that models trained on chaotic and complex (Class III and Class IV) rules focus more on recent states, while models trained on uniform rules exhibit consistently low attention. Periodic rules demonstrate a repeating attention pattern, suggesting that the model is learning to attend to earlier cycles of the same state rather than general state history. **Right:** Average attention across the final 10 states for all rules, plotted against Lempel-Ziv complexity ($r = 0.66$) indicates that models trained on higher complexity data attend more highly to historical states during prediction.

To explore this, we analyze the self-attention scores with respect to the last state in the input sequence, which the model uses to predict the next state (see Section 3.2). Specifically, we examine the attention values corresponding to the final ten states before the target state. Figure 3 illustrates the average attention across all layers and heads for the different ECA rules, as well as the attention at each of the last 10 states for one rule from each complexity class.

Our findings reveal that models trained on rules that produce more complex dynamics tend to allocate higher attention to the last ten states, with a strong positive correlation ($r = 0.66$) between data complexity and average attention. This suggests that models trained on complex data integrate information from past states to make their predictions. In contrast, models trained on simpler rules display lower attention across the last ten states. For example, Rule 168 (uniform) shows consistently low attention, indicating that previous states are not being utilized in the prediction process. Rule 179 (periodic) demonstrates a recurring pattern in its attention scores, where the model focuses on every other state. This behavior is explained by the nature of Rule 179, which produces an alternating pattern that repeats every two time steps (Figure 3). Thus, the model appears to be learning only this alternating cycle, rather than general state history. The simpler attention structure suggests that these models learn a trivial solution, which is in accordance with their poorer downstream performance.

We emphasize that for both rules that generate simple data and those that generate complex data, a trivial solution exists: learning the instantaneous 8-bit rule and applying it to only the current state. Such a model would have no attention on previous states, as it only needs the current state to make its prediction. The fact that the complex models are attending to previous states indicate that they are learning a more complex solution to this simple problem, and we conjecture that this complexity is what makes the model "intelligent" and capable of repurposing learned reasoning to downstream tasks.

We had initially expected that predicting one step in the future would be too easy, and would result in every model learning the trivial (8-bit rule) solution. As such, we trained models to predict 5 steps ahead. Surprisingly, we found that models predicting only the next step not only learned non-trivial solutions, but even outperformed models predicting five steps ahead (Figure 4). This result suggests that even when predicting the immediate next state, the models are learning nontrivial solutions (provided that the underlying data is complex), capable of capturing complex patterns beyond the trivial rule-based approach.

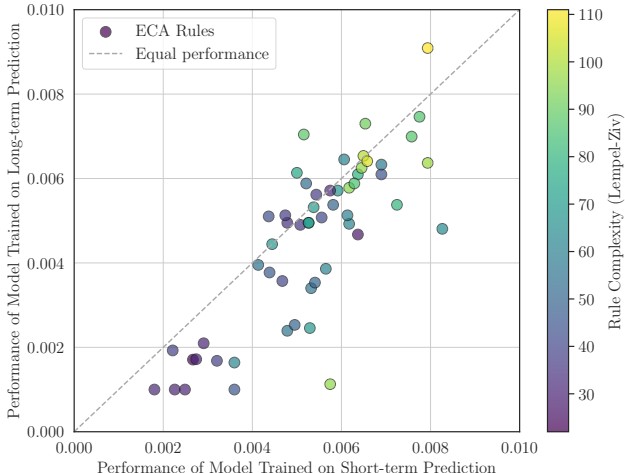

Figure 4: Comparison of model performance on short-term (1-step) and long-term (5-step) prediction tasks for ECA rules. Points are colored by Lempel-Ziv complexity, with the dashed line indicating equal performance. Points below the line show better short-term performance.

In this section, we explored the relationship between system complexity and emergent intelligence. Our results show that downstream model performance improves with pretraining on more complex data, but can deteriorate with excessive complexity or chaotic behavior, signifying a sweet spot of complexity conducive to intelligence. Attention patterns further reveal that models trained on complex data integrate historical information into their predictions, suggesting that they are learning more sophisticated solutions to relatively simple problems. We hypothesize that this complexity in the learned representations is a key factor enabling models to generalize and perform well on downstream tasks.

## 6   RELATED WORK

**Elementary Cellular Automata and Complexity**   Elementary Cellular Automata (ECA) serve as foundational models for exploring complexity arising from simple rules. Early research demonstrated that even minimalistic CA systems, governed by basic and local interaction rules, could generate intricate patterns over time (Wolfram, 1983). Wolfram's extensive investigations into one-dimensional CA revealed that certain rules produce behaviors ranging from stable and periodic to chaotic and complex (Wolfram, 1984). In his work *Cellular Automata and Complexity*, Wolfram classified one-dimensional CA into four classes based on their dynamic behaviors, highlighting the rich complexity that simple systems can exhibit (Wolfram & Mallinckrodt, 1994). Wolfram's *Principle of Computational Equivalence* posits that systems with sufficiently complex behavior can exhibit computational capabilities equivalent to universal computation (Wolfram & Gad-el Hak, 2003). ECA Rule 110, which has been proven to be Turing complete (Cook et al., 2004), demonstrates that even simple, rule-based systems can perform any computation given the right initial conditions. Recent works have explored the use of transformers to predict cellular automata states (Berkovich & Buehler, 2024).

**Computation at the Edge of Chaos**   The concept of computation at the "edge of chaos" suggests that systems poised between order and disorder exhibit maximal computational capabilities and complex behavior. Langton (1990) introduced this idea, demonstrating that cellular automata operating at this critical transition can perform complex computations. Packard (1988) explored how systems adapt toward the edge of chaos, suggesting that evolution may favor systems that balance between stability and chaos. Mitchell et al. (1993) investigated evolving cellular automata to perform computations, finding that rules near the edge of chaos are more capable of complex tasks. These studies provide a theoretical foundation for our work, as we observe that models trained on rules with higher complexity exhibit greater intelligence in downstream tasks.

**Emergence of Intelligence Through Complexity Exposure**    The hypothesis that intelligence can arise from exposure to complexity is supported by studies in artificial life and complexity science. Bedau (2003) discusses how complex behaviors and adaptation emerge from simple rules in artificial life systems. Langton (1990) introduced the concept of "computation at the edge of chaos," proposing that systems poised between order and disorder exhibit maximal computational capabilities and complex behavior. Our conjecture that creating intelligence may require only exposure to complexity aligns with these perspectives.' Crutchfield & Mitchell (1995) explored how evolutionary processes can lead to emergent computation in cellular automata, demonstrating that complexity in the environment can drive the evolution of computational abilities. Kauffman (1992) discussed self-organization and complexity in biological systems, suggesting that complex interactions can lead to emergent properties like intelligence.

**Emergent Abilities in Large Language Models**    Recent advancements in large language models (LLMs) have shown that not only increasing model size but also exposing models to more complex and diverse data can lead to the emergence of new capabilities not present in smaller models or models trained on simpler data. Wei et al. (2022) discuss emergent abilities in LLMs, highlighting that certain reasoning tasks become solvable only when models reach a certain scale and are trained on sufficiently complex data. Brown (2020) demonstrate that LLMs like GPT-3 can perform few-shot learning, indicating that exposure to a wide range of linguistic contexts and complexities enhances the models' adaptability and understanding. Hoffmann et al. (2022) emphasize the importance of data scaling laws, showing that increasing the amount and complexity of training data can lead to better performance than merely increasing model size, suggesting a trade-off between model capacity and data complexity. Kaplan et al. (2020) introduce scaling laws for neural language models, illustrating how performance improves predictably with model size, dataset size, and computational resources, but also noting that the nature of the data plays a crucial role.

**Reservoir Computing and Chaotic Systems**    Reservoir Computing (RC) leverages fixed, high-dimensional dynamical systems—often randomly initialized recurrent neural networks—to model nonlinear and chaotic systems (Maass et al., 2002). It projects input signals into a high-dimensional space, capturing temporal patterns through inherent dynamics while only training the output weights (Lukoševičius & Jaeger, 2009). This simplifies training and allows the system to handle chaotic behavior without adjusting internal weights. RC has been successfully applied to tasks like time series prediction and system identification (Pathak et al., 2018), effectively harnessing chaotic system dynamics for computation.

## 7    DISCUSSION

In this work, we utilize LLMs trained on elementary cellular automata (ECA) to study how intelligent behavior may emerge in large language models (LLMs) when trained on increasingly complex systems. Our findings reveal several important trends that contribute to understanding the relationship between complexity and model behavior.

**Optimal complexity: the "edge of chaos"**    We observe that the best model performance occurs in systems operating at high but not excessive complexity, previously referred to as the "edge of chaos" (Langton, 1990). Models trained on Class IV ECA rules, which exhibit structured yet complex behaviors, perform optimally, suggesting that intelligence may emerge in systems that balance predictability and complexity. On the one hand, if a system is too simple and predictable, like those governed by Class I and II rules, the model quickly learns a trivial solution and fails to develop more sophisticated reasoning. On the other hand, highly chaotic rules from Class III provide too much randomness, akin to training on noise, where the lack of meaningful patterns prevents the model from finding useful structure. The sweet spot arises when complexity is high enough to challenge the model but still retains underlying patterns that the model can exploit. This balance between order and randomness seems particularly conducive to fostering intelligent behavior, as it forces the model to develop more effective reasoning and processing strategies.

**Complex solutions for irreducible systems**    Our findings demonstrate that large models are capable of learning complex, non-trivial solutions even when simpler, more trivial ones are available, likely because they are overparametrized. For ECA rules generating lower-complexity data, the

models often adopt a trivial solution, focusing only on the current state since no history is needed. However, when exposed to more complex data, models tend to leverage prior states, as indicated by the attention patterns in Figure 3. Despite the memoryless nature of ECA systems, overparameterized models explore a broader search space, and solutions that integrate historical information may be more robust. We hypothesize that by learning to incorporate past states, the model develops generalizable logic that can be reused across tasks. In contrast, a model relying solely on a trivial, state-specific rule would struggle to transfer its knowledge to more complex downstream tasks. Thus, the ability to learn from past states may be key to the model's success in adapting to diverse problems.

Certain ECA rules, such as Rule 110, are known to be computationally *irreducible*, meaning their behavior cannot be predicted without directly computing each step (Wolfram, 1997). However, some studies suggest that even in these systems, partial predictability can be achieved under certain conditions (Israeli & Goldenfeld, 2004). This implies that models learning more complex, non-trivial solutions can actually outperform simpler, irreducible approaches by leveraging approximate but efficient predictions. Rather than directly calculating each state, models can explore patterns and generalize from past states, potentially leading to solutions that are not only more robust across tasks but also more efficient than the irreducible solution.

**Broader Impact** Our findings connect to a larger body of work on the emergence of intelligence in large language models (LLMs). Understanding how LLMs develop sophisticated reasoning capabilities when trained on relatively simple data could offer new insights into why and how intelligence emerges in these models. This research may help shed light on some of the open questions surrounding LLMs, particularly how their internal representations evolve and how certain training conditions lead to more transferrable reasoning abilities.

In future work, this framework can be further explored by training larger LLMs on synthetic data generated by simple rule-based systems. Incorporating measures of complexity, such as those used in this study, could provide a valuable tool for prioritizing and curating data, ensuring that models are exposed to information with the right balance of structure and randomness. This aligns with recent advances in data curation, where data quality and complexity, rather than quantity, is increasingly emphasized in improving model performance (Zhao et al., 2023; Cao et al., 2023; Liu et al., 2023).

Additionally, this work may have implications for our understanding of human intelligence, which is proposed to have evolved as a mechanism for interacting with a complex and hard-to-predict world (Euler, 2018). The idea that intelligence arises in systems operating at the "edge of chaos" parallels cognitive science theories suggesting that human brains function at a critical state between different dynamics (Cocchi et al., 2017; Hesse & Gross, 2014; O'Byrne & Jerbi, 2022). By exploring the conditions under which LLMs develop intelligence, we may gain new insights into the fundamental processes that underlie both artificial and human cognition.

## 8 REPRODUCIBILITY STATEMENT

To facilitate the reproduction of our results, we have provided detailed descriptions of our data processing methods and experimental procedures in Sections 3 and 4. We utilized the GPT-2 model obtained from Hugging Face for our experiments. The complete code for our paper is now available in the following GitHub repository: https://github.com/vandijklab/Intelligence_at_the_edge_of_chaos.

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

## A   SCALING EXPERIMENTS

In this section, we explore the interplay between data complexity, model capacity, and data volume in the learning process. We conducted scaling experiments using three transformer-based models:

1. A 67k-parameter custom single-layer transformer with one attention head and 64-dimensional embeddings ("Tiny"),

2. An 85M-parameter GPT-2 small model with 12 layers, 12 attention heads, and 768-dimensional embeddings ("Small"), and

3. A 708M-parameter GPT-2 large model with 36 layers, 20 attention heads, and 1280-dimensional embeddings ("Large").

Each model is pretrained on various quantities of ECA-generated data with varying complexities.

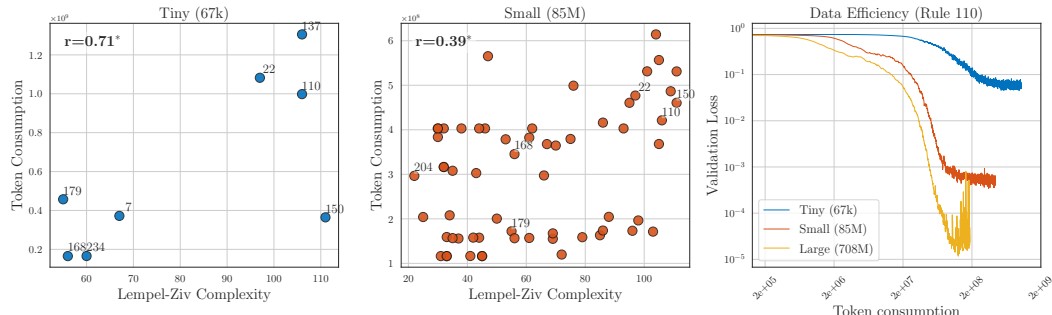

Figure 5: Scaling experiments with varying quantities of data and different model sizes. **Left**: Number of tokens seen before convergence during pre-training for the Tiny model. **Center**: Number of tokens seen before convergence during pretraining for the Small model. **Right**: Validation loss as a function of token consumption for models trained on data from ECA Rule 110. Larger models achieve lower validation loss with fewer tokens, highlighting the improved data efficiency for larger models.

Figure 5 illustrates the results of these experiments. The left and middle panels show that as the complexity of the data increases, both models require more tokens to reach convergence. This trend indicates that higher data complexity demands more data for effective learning. Notably, the right panel demonstrates that the Small model achieves lower validation loss faster than the Tiny model when trained on complex data (and similarly for Large vs. Small), suggesting that larger models can learn more efficiently from complex patterns.

These findings highlight a trade-off between data complexity, model capacity, and data volume. While complex training data can enhance model capabilities, it necessitates more data for smaller models to learn effectively. However, increasing model capacity can mitigate this requirement by enabling faster convergence, thereby reducing the amount of data needed.

## B   THE ROLE OF TEMPORAL STRUCTURE IN LEARNING REPRESENTATIONS

To discern whether temporal structure is necessary or if spatial complexity alone is sufficient for better feature learning, we performed an ablation study by disrupting the sequential nature of the ECA data. Specifically, we randomly shuffled the temporal order of the states generated by the ECAs while preserving their spatial configurations. This manipulation retains the spatial complexity but removes the inherent temporal structure, effectively isolating the spatial component.

We trained models on this temporally shuffled data following the same procedures as detailed in Section 3. The performance of these models on the easy reasoning task showed a significant drop compared to models trained on the original, temporally ordered data. As shown in Figure 6, models trained on temporally structured data reached 80% accuracy in significantly fewer epochs than those trained on temporally shuffled data.

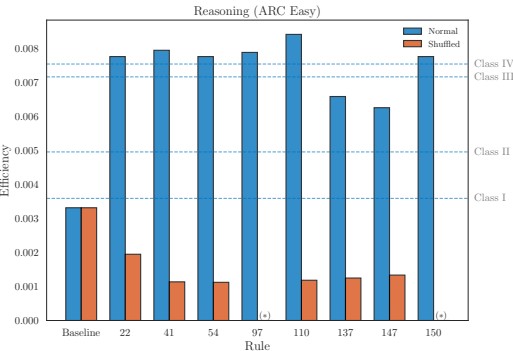

Figure 6: Impact of temporal shuffling on downstream task performance. Models trained on temporally shuffled ECA data exhibit decreased efficiency on the easy reasoning task compared to models trained on ordered data, highlighting the importance of temporal structure in learning effective representations. The average efficiency for each Wolfram class on ordered data is shown with dashed lines for reference. ($*$) indicates models that did not reach the 80% accuracy threshold.

These results underscore the critical role of temporal information in ECA data in enabling models to learn representations that generalize effectively. While spatial complexity provides rich patterns, it is the structured temporal evolution that allows models to capture dynamic relationships and develop more sophisticated reasoning abilities. This experiment confirms that both spatial and temporal complexities are necessary components for training models capable of generalizing and performing complex reasoning tasks.

## C  DOWNSTREAM TASK: NIM GAME

To further evaluate the models' ability to generalize and perform strategic reasoning, we tested their performance on predicting optimal moves in the game of Nim, a classic mathematical strategy game (Bouton, 1901). In Nim, players take turns removing objects from distinct heaps, aiming to be the one who removes the last object. Optimal play involves computing the binary addition of heap sizes, making this game an ideal test for logical reasoning and pattern recognition.

We constructed a dataset comprising sequences of Nim game states and their corresponding optimal moves. To produce diverse scenarios for a challenging task, we simulate games starting with between 5 and 15 heaps and 1 to 10 items in each heap. Each game state is represented as a string starting with the initial configuration of the heaps, followed by a sequence of moves until the game terminates when all the heaps are empty. Each move is encoded as the player number, followed by the heap identifier, followed by the number of items removed. For example, "1B3" signifies "Player 1 removed 3 items from heap B". Each possible move and heap state corresponds to a unique token for the embedding layer of the model. The dataset was split into training and validation sets using a 90-10 ratio.

Using the same approach as in previous downstream tasks, we trained models with the ECA-pretrained weights frozen, allowing only the input embedding and output layers to be updated. Figure 7 shows the relationship between model efficiency and the complexity of the ECA rule used during pretraining. We found that the Nim game is considerably harder than the ARC reasoning tasks, and most models are unable to get accuracies greater than 20%. We adjust our definition of "efficiency" accordingly to 1 divided by number of epochs to reach 20% accuracy. We observed a significant ($p < 0.05$) positive correlation between data complexity and model performance on the Nim task, consistent with our earlier findings on reasoning and chess benchmarks. Models pretrained on more complex data required fewer epochs to reach a given level of accuracy, indicating that complex pretraining data facilitates the acquisition of strategic reasoning abilities.

These results reinforce the idea that pretraining on complex data structures enables models to develop more sophisticated representations, which can be effectively leveraged in downstream tasks.

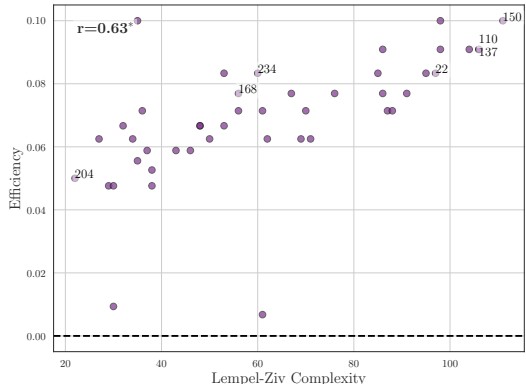

Figure 7: The relationship between performance in the Nim game task and data complexity. Higher data complexity is correlated with increased efficiency, measured as 1 divided by the time to reach 20% accuracy. The data has a statistically significant correlation coefficient of $r = 0.63$, consistent with the trends we observed with the other downstream tasks.

## D    CORRELATION COEFFICIENTS AND HYPOTHESIS TESTING

We computed the Pearson correlation and the distance correlation coefficients (Edelmann et al., 2021) between the rule complexity and the performance on all downstream tasks. Whereas the Pearson correlation coefficient measures the linear dependence between variables, the distance correlation is capable of measuring non-linear association as well. The correlation coefficients and their respective p-values for the Lempel-Ziv complexity are given in Table 1.

Table 1: Summary of Correlation Analysis Results for Downstream Tasks

| Task | Pearson Correlation | Distance Correlation |
|---|---|---|
| ARC-Easy | $0.7322$ ($p = 4.93 \times 10^{-18}$) | $0.7163$ ($p = 4.97 \times 10^{-3}$) |
| ARC-Hard | $0.3896$ ($p = 5.04 \times 10^{-4}$) | $0.4444$ ($p = 4.97 \times 10^{-3}$) |
| Chess | $0.4371$ ($p = 9.73 \times 10^{-3}$) | $0.4329$ ($p = 2.99 \times 10^{-2}$) |
| NIM | $0.6314$ ($p = 3.31 \times 10^{-6}$) | $0.7108$ ($p = 4.97 \times 10^{-3}$) |

## E    RESULTS FOR OTHER COMPLEXITY MEASURES

The downstream results for the ARC Easy, ARC Hard, and Chess tasks as a function of the compression complexity, Lyapunov exponent, and Krylov complexity measures are shown in Figure 8. We observe the same general patterns that we see with the Lempel-Ziv complexity in Figure 2.

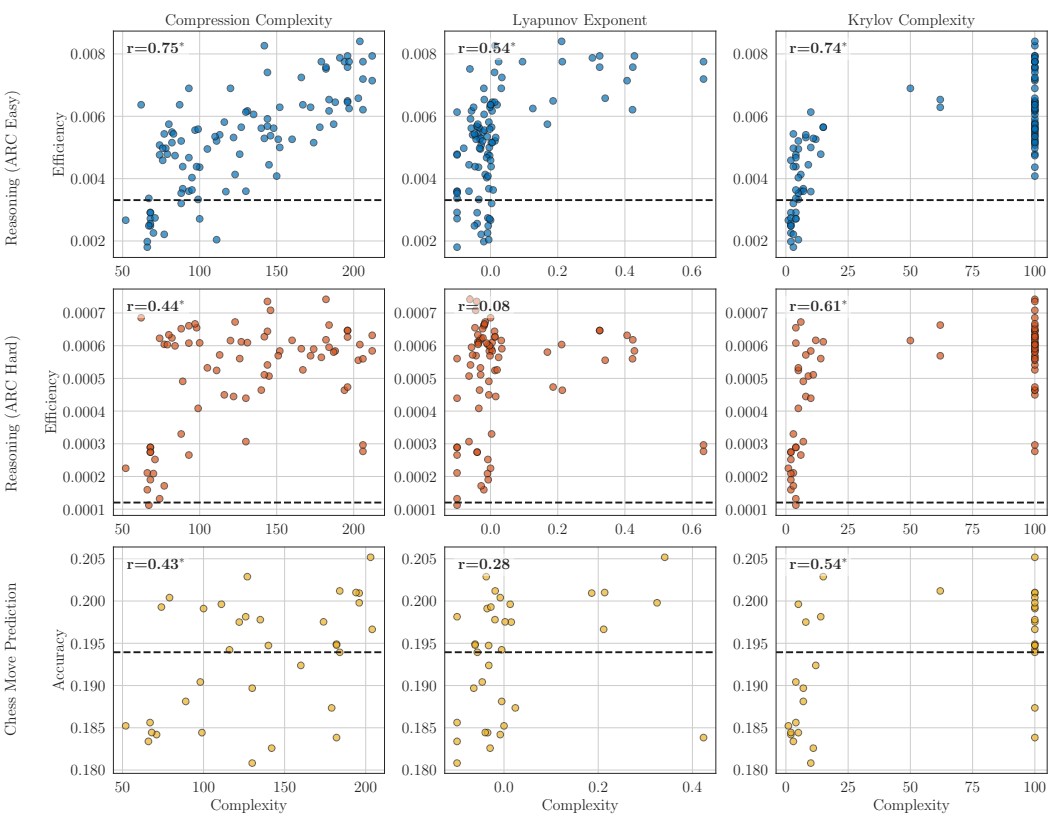

Figure 8: Relationship between downstream task performance and data complexity for other complexity measures. Rows depict easy reasoning, hard reasoning, and chess move prediction tasks, while columns show compression complexity, Lyapunov exponent, and Krylov complexity, respectively. Baseline results for randomly initialized transformers are shown with a dashed black line.

