# OpenReview forum: "Intelligence at the Edge of Chaos"
_ICLR.cc/2025/Conference — ICLR 2025 Poster_

### Official Review · Reviewer_JMh7 · 2024-10-24

**Soundness:** 2
**Presentation:** 3
**Contribution:** 3
**Rating:** 8
**Confidence:** 4

**Summary:**

This paper explores a potential connection between complexity and intelligence using data from elementary cellular automata (ECA) — a class of discrete dynamical systems that exhibit simple, complex, and chaotic dynamics. The authors find that transformers trained on complex and chaotic dynamics perform better on downstream tasks. They first train transformers on these dynamical systems, then adapt the pre-trained transformer to other tasks while freezing the transformer weights and only training the encoder/decoder layers.

**Strengths:**

This paper proposes a novel approach to studying connections between complexity and intelligence. By pre-training transformers on complex patterns, it aligns with large language model (LLM) studies, which have demonstrated that high-quality training data leads to better performance on testing tasks. This study conducts thorough experiments across various tasks, examining the topic from different perspectives. It reveals significant performance differences in transformers trained on different cellular automata.

**Weaknesses:**

The most critical issue in this paper is the method of generating training data. As mentioned in lines 146-152, you first generate a large spatiotemporal evolution, then extract only a small spatiotemporal window from it. This approach inadvertently introduces randomness into the evolution because information outside the spatial window is lost, yet its effects continue to diffuse into the observed space. The authors appear to overlook this randomness, claiming the system is deterministic. They then use a linear encoder/decoder to map states into hidden representations and predict the next state from these hidden states.

This mismatch may lead to several problems, potentially explaining some of the results observed in your paper:

1. Your deterministic model may not be strong enough to learn highly random patterns;
2. The attention pattern discovered in Figure 4 might be caused by this mismatch: For simple tasks, external unknown information may not diffuse into the space, so the cut windows contain truly deterministic dynamics, hence the transformer doesn't need to have attention on previous states. However, for complex patterns, information from states outside the window is necessary for prediction. Consequently, the transformers use previous states for better prediction, resulting in the observed attention patterns;
3. This could lead to a trivial, yet unfalsified explanation of your discovery: the reason transformers trained on complex patterns perform well on downstream tasks might be because they have more attention. And the reason they have more attention might be caused by the unknown information outside the spatial window.

To address this issue, you could predict only part of the next state. For example, given state indices 1-60 as input, you could predict only states 2-59, thereby mitigating the problem. I’m wondering if you still observing the results when removing such randomness.

Some other minor issues:

1. The models should not been named as large language model (LLM) — they are not large, and not natural language. I would suggest using transformer instead;
2. Around line 158, please define “binary vector”. Is it a vector of spatial pattern at given time?
3. In line 232, why there isn’t rule 110 in ARC Easy / Chess Move? It is Turning complete, may need more attention on it.
4. It will be good to compare to a randomly initialized transformer. Would complex ECAs beat random transformer?

**Questions:**

1. How does this relate to Reservoir Computing (RC)? A randomly initialized RC can also perform well on nonlinear and chaotic tasks. How do you distinguish between the sources of good performance—complexity versus reservoir effects?
2. Beyond the trivial explanation mentioned above, what other reasons might explain why the transformer learns to use attention? Have you experimented with varying model sizes? It's possible that more powerful models might not require attention, while smaller models (or those trained with high L2 regularization) might need attention to allocate more computational resources.
3. Does this relate to the principle of computational equivalence?

---

> ### Author Response · Authors · 2024-11-20
>
> Thank you for your detailed review and constructive feedback. We are glad that you found our approach novel and appreciated our use of elementary cellular automata (ECA) as a framework for exploring the connection between complexity and emergent intelligence. In response to your feedback, we have addressed both major and minor points to enhance the robustness of our findings and the clarity of our presentation. Specifically:
>
> * We conducted additional experiments to validate the role of spatiotemporal sampling in our results.
> * Baseline comparisons with randomly initialized transformers were added to evaluate the impact of ECA pretraining.
> * Additional experiments with smaller models were included to assess the role of model capacity in our findings.
>
> Below, we provide detailed responses to your comments and questions.
>
> **Weaknesses**
> **Comment 1:**
> *The method of generating training data introduces randomness due to spatial windowing, which may impact the results. The attention patterns observed might arise from missing information outside the spatial window, rather than intrinsic model behavior.*
> **Response:**
> Great observation\! While it is true that information outside the spatial window could diffuse into the observed space, we suspect that the effect may be minimal for the following reason: at any given state, only two cells (out of 100\) at most could be influenced by this “missing” information, leaving the majority of the state unaffected.
> To validate this, we reran our experiments with models trained on the **entire 1000-dimensional spatial width** to eliminate any randomness. The data was generated with symmetric boundary conditions to ensure there were no boundary effects. We recreated the attention plot from Figure 4 and observed the same patterns as in our original results, as seen [here](https://postimg.cc/62MF4MbH). This suggests that the greater attention observed in complex models is not due to the missing information but rather reflects an intrinsic feature of the models’ behavior.
> Due to resource constraints, we performed this experiment with the four rules shown in Figure 4\. We hope to extend it to all rules for an updated version of the manuscript.
>
> **Comment 2:**
> *The models should not be referred to as “large language models” (LLMs), as they are neither large nor trained on natural language.*
> **Response:**
> We acknowledge this perspective. However, as we are using a GPT-2 architecture ([which is traditionally referred to as an LLM](https://openai.com/index/better-language-models/)), we thought it is fair to use this term prior to pretraining. However, to address your concern, we have updated the manuscript to refer to the trained models as “transformer models” when discussing their post-pretraining behavior.
>
> **Comment 3:**
> *Please clarify the definition of “binary vector” (line 158).*
> **Response:**
> The term “binary vector” was defined in Section 3.1 as representing the spatial pattern of the automaton at a given time. To improve clarity, we have reiterated this definition on line 161\.
>
> **Comment 4:**
> *Why isn’t Rule 110 included in ARC Easy or Chess Move evaluations (line 232)?*
> **Response:**
> At the time of the initial submission, some of our models had not completed training. We have since included Rule 110 in Figure 2 and updated the corresponding analyses.
>
> **Comment 5:**
> *Would complex ECAs outperform a randomly initialized transformer?*
> **Response:**
> To address this, we added baseline results for a randomly initialized transformer. The results show that transformers pretrained on complex ECAs consistently outperform their randomly initialized counterparts, highlighting the value of structured complexity in pretraining. These results have been incorporated into the revised manuscript ([Figure 2](https://postimg.cc/F1QvNkt8)).

---

> > ### Comment · Reviewer_JMh7 · 2024-11-22
> >
> > > Great observation! While it is true that information outside the spatial window could diffuse into the observed space, we suspect that the effect may be minimal for the following reason: at any given state, only two cells (out of 100) at most could be influenced by this “missing” information, leaving the majority of the state unaffected. To validate this, we reran our experiments with models trained on the **entire 1000-dimensional spatial width** to eliminate any randomness. The data was generated with symmetric boundary conditions to ensure there were no boundary effects. We recreated the attention plot from Figure 4 and observed the same patterns as in our original results, as seen [here](https://postimg.cc/62MF4MbH). This suggests that the greater attention observed in complex models is not due to the missing information but rather reflects an intrinsic feature of the models’ behavior.
> > > Due to resource constraints, we performed this experiment with the four rules shown in Figure 4. We hope to extend it to all rules for an updated version of the manuscript.
> >
> > Thank you for implementing the additional experiments. But the updated figure seems lead to more questions:
> >
> > 1. How stable the result is?
> > 2. Why rule 22 reduces its attention on this experiment?
> >
> >
> >
> > > We acknowledge this perspective. However, as we are using a GPT-2 architecture ([which is traditionally referred to as an LLM](https://openai.com/index/better-language-models/)), we thought it is fair to use this term prior to pretraining. However, to address your concern, we have updated the manuscript to refer to the trained models as “transformer models” when discussing their post-pretraining behavior.
> >
> > Thank you for updating the manuscript.
> >
> > > The term “binary vector” was defined in Section 3.1 as representing the spatial pattern of the automaton at a given time. To improve clarity, we have reiterated this definition on line 161.
> >
> > Thank you for clarify this.
> >
> > > To address this, we added baseline results for a randomly initialized transformer. The results show that transformers pretrained on complex ECAs consistently outperform their randomly initialized counterparts, highlighting the value of structured complexity in pretraining. These results have been incorporated into the revised manuscript ([Figure 2](https://postimg.cc/F1QvNkt8)).
> >
> > Thank you for adding this!

---

> > > ### Author Response · Authors · 2024-12-02
> > >
> > > We thank the reviewer for their patience. To provide robust results, we conducted additional experiments on the 1000-width symmetric boundary data, ensuring full spatial context and zero data leakage. As this setup has 10x more information, the models take a significantly longer time to converge, especially for Class III (chaotic) and Class IV (complex) rules. **The results of these new experiments align with our original findings of the relationship between complexity and downstream performance**, reinforcing the validity of our conclusions.
> > >
> > > ## Robustness of Downstream Performance
> > > Evaluating the 1000-width models on the ARC Hard downstream task confirmed that the trends observed with 100-width models remain consistent. Importantly, **the correlation between complexity and downstream performance persisted, with a Pearson correlation of 0.48 (p = 0.0229) ([Figure link](https://postimg.cc/D8Vw5v6B))**. These results further support our hypothesis that structured complexity fosters generalization.
> > >
> > > ## Attention Patterns
> > > We expanded our analysis of the attention patterns of 1000-width models to include a wider range of rules. Chaotic rules (e.g., Rule 22) exhibited greater variability in their attention patterns, likely due to the unpredictable nature of the data. However, the overall trend remained consistent: **models trained on more complex rules allocated greater attention to recent states**. This was corroborated by a **significant positive correlation between complexity and average attention (r = 0.56, p = 0.0079, [Figure link](https://postimg.cc/JHCC3n8X))**.
> > >
> > > ## Conclusion
> > > These comprehensive experiments demonstrate that the observed trends are intrinsic to the models’ pretraining dynamics and not artifacts of the sampling method. We hope these results satisfactorily address the reviewer’s concerns, and ask that they consider these results in their evaluation of our work.

---

> > > > ### Comment · Reviewer_JMh7 · 2024-12-02
> > > >
> > > > Thank you very much! This new result is very promising and address all my concerns (data leak, significance, etc.). I will increase my rating accordingly.

---

> ### Author Response · Authors · 2024-11-20
>
> **Question 1:**
> *How does this relate to Reservoir Computing (RC)?*
> **Response:**
> Thank you for this insightful question. While reservoir computing (RC) leverages the fixed dynamics of randomly initialized reservoirs to solve tasks, our study fully trains transformer model parameters on specific ECA rules. This allows our models to adapt to the complexity of the data, whereas RC relies on the reservoir’s inherent dynamics.
> Our findings highlight that models trained on complex rules develop **richer internal representations**, as evidenced by their attention patterns. This suggests that improvements in performance are driven by the **training data complexity**, not fixed dynamics, distinguishing our work from traditional RC approaches. We have added a section to the related work highlighting these comparisons.
>
> **Question 2:**
> *Beyond the trivial explanation mentioned above, what other reasons might explain why the transformer learns to use attention? Have you experimented with varying model sizes?*
> **Response:**
> Beyond external randomness, the transformer’s use of attention likely reflects its capacity to model long-range dependencies in complex data. To investigate this further, we conducted an additional experiment with a **small, one-layer transformer model** (Appendix A). Our findings show that smaller models require more data to achieve comparable performance, suggesting that attention use is influenced by both the complexity of the data and the model’s capacity. We plan to explore scaling laws with larger models in future work.
>
> **Question 3:**
> *Does this relate to the principle of computational equivalence?*
> **Response:**
> Yes, this concept is highly relevant to our work\! The principle of computational equivalence suggests that processes that are not trivially simple are capable of similar levels of computation. ECAs are a canonical example of this, as Wolfram used them to illustrate the principle.
> However, our findings suggest an important nuance: while chaotic and complex systems may both be computationally equivalent, models trained on chaotic data may require more data or compute to learn effective representations compared to models trained on complex data. This aligns with the idea that the efficiency of computation differs across systems.
> We have added a discussion of this principle to the **Related Work** section of the manuscript.
>
> We hope the revisions address your concerns and improve the overall quality of the manuscript.

---

### Official Review · Reviewer_8JtY · 2024-10-28

**Soundness:** 3
**Presentation:** 3
**Contribution:** 3
**Rating:** 6
**Confidence:** 3

**Summary:**

The authors of this paper utilize a large language model (GPT-2) to demonstrate "Intelligence at the Edge of Chaos," showing that models achieve optimal performance in moderately complex, but not overly chaotic, systems. They train on tasks based on elementary cellular automata (ECA) and evaluate on various downstream tasks to explore the link between system complexity and emergent intelligence. The results align well with their hypothesis, though additional experiments could further strengthen these findings.

**Strengths:**

The paper’s topic is innovative, as I believe the emergence of intelligence is fostered by increasingly large architectures capable of handling complex data. The authors devote significant attention to explaining the concept of the "Edge of Chaos," ensuring a thorough understanding of this idea. Their explanation of the results is also convincing, and the paper’s visual illustrations are highly effective.

**Weaknesses:**

The experiments are somewhat limited. Adding more downstream tasks, such as nim games or toroidal chess, could enhance the findings. Additionally, testing the hypothesis on smaller models could provide further support, rather than relying solely on results from larger models.

**Questions:**

1. Could you include more downstream tasks, like nim games or cylinder chess, to expand the evaluation?

2. Could you test on smaller models, such as ResNet-50, BERT, or Mamba-170m, to validate the results?

---

> ### Author Response · Authors · 2024-11-20
>
> Thank you for your review and thoughtful feedback. We are pleased that you found our work innovative and appreciated our focus on the “Edge of Chaos” concept and its connection to emergent intelligence. In response to your feedback, we have taken steps to address the identified weaknesses and expand the scope of our evaluation.
>
> **Question 1:**
> *Could you include more downstream tasks, like Nim games or cylinder chess, to expand the evaluation?*
>
> **Response:**
> Great suggestion! To expand the evaluation, we included a downstream task based on the **Nim game**. The results show a trend consistent with our other downstream tasks, where models pretrained on higher-complexity CA data perform better on the Nim game task. This further supports our hypothesis that structured complexity fosters better generalization.
> We have incorporated the details and results of this experiment into [Figure 8](https://postimg.cc/ZWfwhy3t) in Appendix C in the revised manuscript.
>
> **Question 2:**
> *Could you test on smaller models, such as ResNet-50, BERT, or Mamba-170m, to validate the results?*
>
> **Response:**
> To evaluate the impact of model size, we conducted scaling law experiments with different model sizes, including the GPT-2 small (85M parameters) model used in our other experiments and also a new one-layer transformer (67k parameters) model. These experiments confirmed our findings, which are included in Appendix A:
> - Models trained on higher-complexity CA data require more training data to converge compared to simpler data ([Figure 6](https://postimg.cc/DSDYXXzQ)).
> - Larger models pretrained on the same CA rule achieve equivalent validation performance with less data, highlighting a **data efficiency advantage for larger architectures**.
>
> These results suggest that both the **complexity of the training data** and the **capacity of the model** influence the trade-off between data volume and performance. We plan to explore this scaling behavior further in future work.
>
> We hope these additions address your concerns and strengthen the overall contribution of our work.

---

> > ### Comment · Reviewer_8JtY · 2024-11-24
> > **Official Comment by Reviewer 8JtY**
> >
> > I believe that conducting additional experiments with more challenging tests and larger models could better illustrate your viewpoint. Nonetheless, I maintain a positive outlook on your paper.

---

> > > ### Author Response · Authors · 2024-11-25
> > >
> > > Thank you for your feedback and positive outlook on our work. For the downstream tasks, we believe the addition of the Nim game provides a well-rounded evaluation on 4 diverse downstream tasks, spanning a spectrum of difficulties—from the easy and hard reasoning tasks to the highly challenging chess task.
> > >
> > > We agree that additional experiments with larger models could provide valuable insights. Based on your suggestion, we conducted further experiments with a **GPT-2 Large (708M parameter) model** on the ARC Hard downstream task. These experiments revealed the same trend observed with the GPT-2 Small model: **pretraining on higher-complexity ECA data leads to better downstream performance**, with a correlation coefficient of 0.66 ([figure](https://postimg.cc/LJbQYKy6)). To strengthen these findings, we are planning to add additional data points to further validate the observed trends. We will include these additional experiments in the final manuscript.

---

### Official Review · Reviewer_YCfn · 2024-10-31

**Soundness:** 2
**Presentation:** 3
**Contribution:** 3
**Rating:** 6
**Confidence:** 3

**Summary:**

The authors explore how the complexity of the rule of the data generative system influences the model capabilities on the downstream tasks. They use elementary cellular automata (ECA) as a rule-based dynamic system to control complexity of the training data trained GPT-2 derived transformer architecture and evaluated the trained model on reasoning/chess move prediction tasks. They found that higher but not too chaotic complex system leads to higher downstream task performance. Also, they found that the model trained with more complex system tends to obtain more complex solution rather than a simple rule, which the authors hypothesized as a reason for better downstream task performance.

**Strengths:**

- I really enjoyed reading the paper. The authors explore the impact of rule complexity of the training data on the model's downstream task performance. I believe it is important and timely subject as there are already many literatures suggesting the importance of data property on acquired intelligence of the neural network models.
- The paper is well motivated and the authors made good connection to the related previous works on complexity and computational capabilities.
- Particularly, I liked the ECA data model that the authors use, which provides a good control of complexity of the rule-based dynamical system. Also, the authors used multiple measures of complexity rather than single definition.
- The paper is well-written, it was easy to follow and understand the key claims that the authors make.

**Weaknesses:**

- While I am fairly convinced with the motivation and general results, but the presented quantitative result is rather weak in the results and figures they provide. Is the difference between accuracy across complexity, for example, in figure 2, shows fairly low r value.
For wolfram's complexity class categorization, does the p-test shows significance? The absolute accuracy difference between the categories look fairly low. And the baseline performance without the ECA pretraining is not provided.

- Did the authors run experiments with multiple seeds?

- The claimed 'emergent' intelligence is only evaluated on the downstream tasks after post-tuning, and in limited kinds. Also, the baseline performance without the pre-training on ECA is not given.

- While authors found that the model trained with complex system tends to obtain more complex solution than the actual rule and made a interesting conjecture that this might be the reason for better downstream performance, no rigorous experiment to further investigate this point is provided.

Further questions and comments are given in the next section.

**Questions:**

- Could you comment on the differences between "complexity" and "diversity" in ECA model? Is any of the complexity measures you incorporate captures diversity of the data as well? Are complex data always more diverse and vice-versa?

-  In figure 2 violin plot, it seems like it's not only mean values but also variances that are correlated to the complexity measure and downstream tasks which I found interesting.  Could the authors provide comment?

- Related to the weaknesses I wrote above, I am curious about baseline performance of the models before pre-trained on ECA data.

- I see different tendency on Krylov complexity compare to the other on fig 3 i.e. high variance on both very low and very high complexity measure. Is there any reason that you could think of?

---

> ### Author Response · Authors · 2024-11-20
>
> Thank you for your thoughtful and detailed review. We are delighted that you found our paper well-motivated and enjoyed the use of elementary cellular automata (ECA) as a framework for controlling and analyzing complexity. In response to your feedback, we have included the baseline performance of untrained models in the relevant figures (e.g., Figure 2\) to provide a clearer comparison. Below, we provide detailed responses to your questions and comments.
>
> **Weakness 1:**
> *The presented quantitative results are rather weak in terms of significance and correlation values (e.g., Figure 2 shows low r-values).*
> **Response:**
> We acknowledge that the correlation coefficient in Figure 2 is not particularly high. However, we hypothesized that this would occur because the relationship between complexity and performance is **non-monotonic**, and this is in fact what we observe: performance peaks at intermediate complexity levels (the “edge of chaos”) and decreases at both lower and higher complexities. This non-linear trend inherently limits the strength of the linear correlation coefficient as a descriptor.
> In order to quantify the nonlinear association, we computed the distance correlation coefficient, a measure of nonlinear correlation, and found that it is statistically significant for each of our downstream tasks as well. The distance correlation coefficients were **0.7163 for ARC Easy**, **0.4444 for ARC Hard**, and **0.4329 for Chess.** We will add these to the manuscript.
>
> To avoid misunderstanding, we clarified in the manuscript that the correlation coefficient is provided to quantify the overall relationship, but the **key takeaway is the qualitative insight**: performance peaks at intermediate complexity and deteriorates with excessive complexity.
>
> **Weakness 2:**
> *For Wolfram’s complexity class categorization, does the p-test show significance? The absolute accuracy difference between categories seems low.*
> **Response:**
> We performed significance tests to analyze the significance of differences between Wolfram’s complexity classes.
>
> * For the **ARC-E and ARC-H tasks**, differences between all categories are statistically significant except for **Class III vs. Class IV**, which aligns with expectations, as both classes produce highly complex data and only sufficiently chaotic rules deteriorate in performance.
> * For the **chess task**, both the Pearson correlation and the distance correlation coefficient were statistically significant. However, we observed that differences between Wolfram classes were not significant, likely due to the smaller number of rules sampled from each class (owing to the high computational cost of training chess models).
>
> We will add these significance tests to the manuscript.
>
> **Weakness 3:**
> *Did the authors run experiments with multiple seeds?*
> **Response:**
> Due to resource constraints, we were unable to repeat the full training procedure for all models across multiple seeds. However, we conducted runs for several models with different seeds and observed **nearly identical training dynamics**, suggesting that the results are consistent across seeds. The training runs can be seen [here](https://postimg.cc/5XXqqn7H). We plan to add additional replicates in an updated version of the manuscript.

---

> ### Author Response · Authors · 2024-11-20
>
> **Question 1:**
> *Could you comment on the differences between “complexity” and “diversity” in the ECA model? Do any of the complexity measures capture diversity as well? Are complex data always more diverse and vice versa?*
> **Response:**
> Complexity and diversity, while interrelated, capture different characteristics of data generated by ECA models. **Complexity** refers to the intricacy of the patterns and structures observed over time, reflecting how challenging the underlying system is to predict or describe. It often emerges from the interplay between order and chaos, as seen in systems operating at the "Edge of Chaos." **Diversity**, on the other hand, pertains to the richness or variety of states and transitions that a system can exhibit. It focuses on the breadth of possibilities rather than the difficulty of understanding or predicting them.
>
> Complex systems are not necessarily diverse, and diverse systems are not always complex. For example:
> - A highly chaotic ECA rule may produce a narrow range of unpredictable states (high complexity, low diversity).
> - Conversely, a periodic ECA rule may exhibit many unique but easily predictable states (low complexity, high diversity).
>
> Some measures of complexity, like **Krylov complexity**, partially overlap with diversity by capturing the number of unique states observed. However, the relationship is not direct or universal across all complexity measures, as many focus on structural or temporal patterns rather than state variety. We believe this distinction is critical to understanding how ECA-generated data influences model performance.
>
> **Question 2:**
> *In Figure 2 (violin plot), it seems like variance is also correlated with complexity and downstream tasks. Could the authors comment on this?*
> **Response:**
> We agree that the variance in performance is an interesting observation. The variance is mostly explained with the **number of distinct rules in each complexity class**. For example, periodic rules (Class II) encompass a broader range of distinct rules than other classes, leading to greater variability in the generated data and model performance.
>
> **Question 3:**
> *I see a different tendency for Krylov complexity compared to the other measures in Figure 3\. High variance is observed for both low and high Krylov complexity. Why is this?*
> **Response:**
> Krylov complexity measures the number of linearly independent basis vectors generated by a dynamical system’s evolution operator. Its behavior is distinct from other measures because:
>
> 1. It is **bounded by the dimensionality of the system**.
> 2. For simple or periodic states, Krylov complexity remains low as these systems evolve in small subspaces.
> 3. For complex and chaotic systems in finite-dimensional space, Krylov complexity saturates quickly as they span the full space \[1\].
>
> This explains the observed behavior in Figure 3: rules with simple or periodic dynamics exhibit low Krylov complexity, while chaotic and complex rules show high Krylov complexity with little middle ground.
> This distinction underscores why we report results with five different complexity measures (Lempel-Ziv, Krylov, compression complexity, Lyapunov exponent, and Wolfram class), as each captures unique aspects of the system’s dynamics.
>
> **Question 4:**
> *Baseline performance before pretraining on ECA data is not provided.*
> **Response:**
> We have now added baseline performance using randomly initialized transformer models in the relevant figures (e.g., [Figure 2](https://postimg.cc/F1QvNkt8)). These results clearly demonstrate the improvements achieved through ECA pretraining.
>
> We hope the revisions address your concerns and enhance the overall quality of the manuscript.
>
> **References**
> \[1\] Hashimoto, Koji, et al. "Krylov complexity and chaos in quantum mechanics." Journal of High Energy Physics 2023.11 (2023): 1-41.

---

> ### Comment · Reviewer_YCfn · 2024-11-25
>
> I appreciate the authors response and additional experiments and analysis.
> I still think the evaluation and analysis on downstream tasks are rather limited, though I think it is a interesting setup.
> I will keep my score as it is, looking forward for the future work.

---

> > ### Author Response · Authors · 2024-11-25
> >
> > Thank you for your feedback and for recognizing the value of our setup. To expand our downstream analysis, we introduced a Nim game task to further validate our findings ([Figure 8](https://postimg.cc/ZWfwhy3t) in Appendix C). The downstream performance on this task, as a function of complexity, showed a statistically significant correlation coefficient of 0.63, consistent with the trends observed in our other downstream tasks. We hope this addition strengthens the evaluation and addresses your concerns about the scope of our analysis.

---

### Official Review · Reviewer_rE2s · 2024-11-03

**Soundness:** 3
**Presentation:** 2
**Contribution:** 3
**Rating:** 8
**Confidence:** 4

**Summary:**

The paper explores whether LLMs can develop generalisable reasoning abilities by pretraining on structured but simple datasets, specifically sequences generated by cellular automata. The authors hypothesise that exposing models to complex patterns, even if the underlying system is deterministic and rule-based, could improve the models’ performance on reasoning tasks. By pretraining LLMs on cellular automata generated data, they test whether this training improves performance on abstract reasoning benchmarks, such as tasks from the Abstraction and Reasoning Corpus and chess move prediction (after finetuning).

The results show that models exposed to complex patterns, those generated by "edge-of-chaos" traces, perform better on these reasoning tasks compared to models pretrained on simpler or more chaotic traces. This suggests that pretraining on richer, more complex patterns helps LLMs develop broader generalisation abilities, even when the training data lacks semantic content.

**Strengths:**

I like about this paper that it uses a relatively simple approach to create data with varying (and quantifyiable) degrees of complexity. The approach could help in exposing LLMs to large amounts of data, potentially leading to smaller requireemtns of corpora during pretraining. The "edge-of-chaos" idea has been investigated in neural networks before, but not as a result of pretraining with data from CAs.

The approach of pretraining on CAs and finetuning the outputs later seems like a great fit to learning representations / ICLR - an interesting way to derive good features for training models.

**Weaknesses:**

While I like the idea and the approach of the paper, the choice of words could be improved and language be more precise. The choice of "Intelligence" in the title and throughout the paper is awkward, the work doesn’t necessarily show "intelligence" in the sense of true reasoning or understanding: it demonstrates however that structured complexity in data can lead to the development of generalisable features or representations, that is a better choice would be to describe the approach as learning useful features or representations for predictions or other complex tasks.

The language around the CA rules is also imprecise - all CA rules are similarly simple, but some of the rules lead to complex (or chaotic, or simple...) outputs / traces. It is the generated data that exhibits the complex patterns in case of the class iv CAs.

As an ablation, it might also be interesting to explore if it is the *structured* complexity that leads to better features, or if complex patterns are sufficient. This could be achieved by using surrogate outputs from CAs (ie scramble the temporal order), to allow the distinction between the complex (spatial) pattern or the structured, sequential nature of the data.

**Questions:**

In my view the paper is quite interesting and a good fit for ICLR, and would like see it accepted if some of its weaknesses are addressed. Some questions:

- (see above) What role does the temporal structure of CA sequences play in learning generalisable representations?
- Is the amount of data required from CA pretraining comparable to the amount of data required using natural langage (or similar complex data), to reach similar generalisation effects?
- what's the trade-off between complexity and volume of data

---

> ### Author Response · Authors · 2024-11-20
>
> Thank you for your thoughtful review and constructive feedback. We are glad that the reviewer found our work interesting and appreciated the simplicity of our approach to generating structured data with varying degrees of complexity. We also appreciate your recognition of the novel use of cellular automata (CA) data to pretrain large language models (LLMs).
>
> We agree that improving the precision of our language and addressing ambiguities would enhance the clarity and impact of our work. In response to your comments, we have made the following high-level changes:
> - We have limited the use of the term “intelligence” and instead emphasized the development of structured and generalizable representations throughout the manuscript.
> - We have clarified that all CA rules are simple in their definitions, but their outputs differ in complexity. Where applicable, we explicitly refer to the complexity of the generated data rather than the rules themselves.
> - Added experiments: We incorporated new results to explore the role of temporal structure in learning representations, as well as the relationship between data volume and model performance as suggested.
>
> Below, we address each of your specific questions in detail.
>
> **Question 1:**
> *What role does the temporal structure of CA sequences play in learning generalizable representations?*
>
> **Response:**
> To investigate this, we conducted the suggested experiment by randomly shuffling the temporal order of elementary cellular automata (ECA) states during training while preserving their spatial organization. This augmentation thus removes the sequential information of the data, isolating the spatial patterns.
> Our findings show that this operation resulted in a **significant drop in downstream performance (ARC Easy)**. This demonstrates that **temporal structure is critical** for learning representations that generalize effectively. Spatial complexity alone is insufficient to account for the observed improvements; temporal information plays a key role in enabling models to learn more sophisticated representations. We have incorporated the results of this experiment into [Figure 7](https://postimg.cc/vgjR8NVy) in Appendix B in the revised manuscript.
>
> **Questions 2 and 3:**
> *Is the amount of data required from CA pretraining comparable to the amount of data required using natural language (or similar complex data) to reach similar generalization effects? What is the trade-off between complexity and volume of data?*
>
> **Response:**
> Interesting questions! Comparing data requirements between cellular automata (CA) and natural language is inherently challenging. While CA-generated data can exhibit high structural complexity, natural language is likely more **information-dense**, embedding semantic relationships, contextual nuances, and real-world grounding. These attributes likely contribute to natural language’s capacity to foster generalization across diverse tasks. However, **quantifying this information density** is challenging due to the lack of universal complexity metrics for natural language, unlike the well-defined measures available for CAs.
> To address this rigorously, we envision an experiment where models are pretrained on a fixed amount of natural language data and evaluated in the same manner as the ECA-pretrained models, e.g. on ARC, chess, etc. By controlling the number of tokens seen during pretraining, we could compare relative performance. However, carrying out such an experiment as well as measuring the complexity of natural language is beyond the scope of this study.
>
> As an alternative approach to answer these questions that is within the scope of our work, we performed a new experiment to investigate the **trade-off between complexity and data volume** by analyzing model performance as a function of the number of tokens seen during ECA pre-training (see Appendix A). We found that **models trained on higher-complexity ECA data require more training data to converge** compared to simpler data ([Figure 6](https://postimg.cc/DSDYXXzQ)). Additionally, **larger models pre-trained on the same ECA rule achieve equivalent validation performance with less data** (and converge to better performance overall), highlighting a data efficiency advantage for larger models.
> These results suggest that both the complexity of the training data and the capacity of the model significantly influence the trade-off between data volume and performance. We have included these findings in Figure 6 and plan to explore the scaling behavior in more detail in future work.

---

> > ### Comment · Reviewer_rE2s · 2024-11-26
> >
> > Thank you for your responses.
> > I'm positive about the changes to the submission, the answers, and the additional implementations in short time, and would be happy to see the paper accepted. I have raised my score to reflect this.

---

### Author Response · Authors · 2024-11-20

We sincerely thank all the reviewers for their insightful comments and are encouraged by their unanimous approval of our work.
\[rE2s\] and \[YCfn\] appreciated the innovative use of elementary cellular automata (ECA) as a framework for controlling and analyzing complexity in pretraining transformer models. \[8JtY\] and \[JMh7\] valued our focus on the "Edge of Chaos" concept and its connection to emergent intelligence, noting the novelty and potential of our approach.

Several reviewers raised important points that we have addressed:

* **Baseline Comparisons**: **\[rE2s\]**, **\[YCfn\]**, and **\[JMh7\]** requested comparisons with randomly initialized transformers. We have now included baseline performance results, demonstrating that models pretrained on complex ECA data outperform their untrained counterparts.
* **Additional Experiments**: **\[rE2s\]** and **\[8JtY\]** suggested exploring the role of temporal structure and incorporating more downstream tasks. We conducted new experiments that highlight the critical role of temporal dynamics in learning generalizable representations and added a downstream task based on the Nim game, which reinforces our findings.
* **Model Size and Capacity**: **\[8JtY\]** and **\[JMh7\]** recommended testing with different model sizes. We performed experiments with smaller transformer models to assess the impact of model capacity on performance, confirming that both data complexity and model size influence learning efficiency.
* **Complexity Measures and Statistical Significance**: **\[YCfn\]** and **\[JMh7\]** questioned the statistical significance of our results and the measures of complexity used. We have clarified our analysis, provided additional statistical tests, and expanded our discussion on different complexity measures to address these concerns.
* **Theoretical Connections**: **\[JMh7\]** inquired about the relation to Reservoir Computing and the principle of computational equivalence. We have added discussions to situate our work within these theoretical frameworks, emphasizing how our findings offer new insights into the computational capabilities of neural networks trained on complex data.

We once again thank the reviewers for their valuable feedback, which has significantly strengthened our paper. We are excited about the future implications of this work and look forward to further advancements in understanding the interplay between data complexity and emergent behavior in machine learning models.

---

### Author Response · Authors · 2024-12-02

We appreciate all of the thoughtful feedback and constructive comments throughout this discussion. Below, we summarize the additional experiments conducted to address the reviewers’ questions:
- **Temporal Structure:** We examined the impact of temporal structure by shuffling the order of ECA states during training. The results confirmed that temporal structure is critical for learning effective representations (and not spatial complexity alone), as its removal led to significant drops in downstream performance. ([Shuffled vs non-shuffled](https://postimg.cc/vgjR8NVy))
- **Complexity vs Data Volume:** We analyzed the interplay between data complexity, data volume, and model size. Our findings showed that higher-complexity data requires more training tokens to converge, and larger models exhibit greater data efficiency. These results provide insights into the relationship between data complexity and model capacity, highlighting it as an important direction for future research. ([Token consumption](https://postimg.cc/DSDYXXzQ), [large model downstream performance](https://postimg.cc/LJbQYKy6))
- **Additional Downstream Tasks:** We introduced the Nim game as a new downstream task to expand our evaluation, further validating that models pre-trained on higher complexity ECA data achieve superior generalization. ([Nim game performance](https://postimg.cc/ZWfwhy3t))
- **Full spatial context:** We validated that the trend of improved downstream performance persists when using the full spatial context, ruling out spatial sampling as a confounding factor. ([Downstream performance of full spatial model](https://postimg.cc/D8Vw5v6B))

In addition to these experiments, we have addressed numerous questions, providing clarifications on key aspects such as the distinction between complexity and diversity, baseline performance comparisons, and connections to related topics like reservoir computing and the principle of computational equivalence. We believe these additions significantly strengthen our work and hope that the reviewers will consider them in their evaluation.

---

### Meta-Review · Area_Chair_RjXh · 2024-12-21

**Metareview:**

This paper investigates whether transformers, in particular GPT-2, can learn to predict simple cellular automata, and also how they do it. This is an unexpected and quite interesting topic, and the investigation is well executed. The uncommon topic/perspective makes it worthy of spotlighting, IMHO. There are no particular weaknesses.

**Additional Comments On Reviewer Discussion:**

The authors have responded well to concerns from the reviewers, including running additional experiments when needed.

---

> ### Public Comment · ~Damien_Teney1 · 2025-04-06
> **Missing related work**
>
> This is a great paper, but it's missing a discussion of the existing literature on the pretraining of transformers on procedural data.
>
> Some examples below, covering language/reasoning data [1-11] and vision data [12].
>
> 1. [Insights into Pre-training via Simpler Synthetic Tasks (2022)](https://arxiv.org/pdf/2206.10139.pdf)
> 2. [Pretraining with Artificial Language: Studying Transferable Knowledge in LMs (2022)](https://arxiv.org/pdf/2203.10326.pdf)
> 3. [On the Transferability of Pre-trained Language Models: A Study from Artificial Datasets (2022)](https://arxiv.org/pdf/2109.03537.pdf)
> 4. [Injecting Structural Hints: Using Language Models to Study Inductive Biases in Language Learning (2023)](https://arxiv.org/pdf/2304.13060.pdf)
> 5. [Modeling Rapid Language Learning by Distilling Bayesian Priors into ANNs (2023)](https://arxiv.org/pdf/2305.14701.pdf)
> 6. [Synthetic Pre-Training Tasks for Neural Machine Translation (2023)](https://aclanthology.org/2023.findings-acl.512.pdf)
> 7. [Exploring Model Depth and Data Complexity Through the Lens of Cellular Automata (2024)](https://openreview.net/pdf?id=SGoI97b5KK)
> 8. [Learning Universal Predictors (2024)](https://arxiv.org/pdf/2401.14953.pdf)
> 9. [Pre-training with Synthetic Data Helps Offline Reinforcement Learning (2024)](https://arxiv.org/pdf/2310.00771.pdf)
> 10. [SIP: Injecting a Structural Inductive Bias into a Seq2Seq Model by Simulation (2024)](https://arxiv.org/pdf/2310.00796.pdf)
> 11. [Between Circuits and Chomsky: Pre-pretraining on Formal Languages Imparts Linguistic Biases (2025)](https://arxiv.org/pdf/2502.19249.pdf)
> 12. [Scaling Backwards: Minimal Synthetic Pre-training? (2024)](https://arxiv.org/pdf/2408.00677.pdf)
>
> ---
>
> For future readers of this paper, it may also be worth highlighting that the "improvements" when pretraining on ECA data are truly marginal. On the chess move prediction, things improve, at best, from 19.5 to 20.5% accuracy. On the other tailor-made tasks ("ARC easy/hard"), there's no improvement in accuracy, only a faster convergence.
>
> BTW the summary given in the meta review above completely misses the point of the paper.

---

### Decision · Program_Chairs · 2025-01-22

Accept (Poster)